# Sex-Dependent Gut Microbiota Features and Functional Signatures in Metabolic Disfunction-Associated Steatotic Liver Disease

**DOI:** 10.3390/nu16234198

**Published:** 2024-12-04

**Authors:** Paola Mogna-Peláez, José I. Riezu-Boj, Fermin I. Milagro, Iñigo Clemente-Larramendi, Sergio Esteban Echeverría, José I. Herrero, Mariana Elorz, Alberto Benito-Boillos, Ana Luz Tobaruela-Resola, Pedro González-Muniesa, Josep A. Tur, J. Alfredo Martínez, Itziar Abete, M. Angeles Zulet

**Affiliations:** 1Department of Nutrition, Food Sciences and Physiology and Centre for Nutrition Research, Faculty of Pharmacy and Nutrition, University of Navarra, 31008 Pamplona, Spain; pmogna@unav.es (P.M.-P.); jiriezu@unav.es (J.I.R.-B.); fmilagro@unav.es (F.I.M.); iclemente.1@alumni.unav.es (I.C.-L.); sergioestebanecheverria@gmail.com (S.E.E.); atobaruela@unav.es (A.L.T.-R.); pgonmun@unav.es (P.G.-M.); iabetego@unav.es (I.A.); 2Navarra Institute for Health Research (IdiSNA), 31008 Pamplona, Spain; iherrero@unav.es (J.I.H.); marelorz@unav.es (M.E.); albenitob@unav.es (A.B.-B.); 3Biomedical Research Centre Network in Physiopathology of Obesity and Nutrition (CIBERobn), Instituto de Salud Carlos III, 28029 Madrid, Spain; pep.tur@uib.es (J.A.T.); jalfmtz@unav.es (J.A.M.); 4Liver Unit, Clínica Universidad de Navarra, 31008 Pamplona, Spain; 5Biomedical Research Centre Network in Hepatic and Digestive Diseases (CIBERehd), 28029 Madrid, Spain; 6Department of Radiology, Clínica Universidad de Navarra, 31008 Pamplona, Spain; 7Research Group on Community Nutrition and Oxidative Stress, University of Balearic Islands-IUNICS & IDISBA, 07122 Palma, Spain; 8Precision Nutrition and Cardiovascular Health Program, IMDEA Food, CEI UAM + CSIC, 28049 Madrid, Spain

**Keywords:** dysbiosis, inflammation, insulin resistance, metabolic syndrome, obesity, personalized medicine

## Abstract

**Background/Objectives**: This study investigates the gut microbiota’s role in metabolic dysfunction-associated steatotic liver disease (MASLD), focusing on microbial and functional signatures and sex-based differences. **Methods**: Using baseline data from 98 MASLD patients and 45 controls from the Fatty Liver in Obesity (FLiO) study, the gut microbiota was profiled with 16S gene sequencing, followed by statistical and machine learning analyses to identify disease-associated microbial signatures. **Results**: Notable alpha and beta diversity differences were observed between MASLD patients and the controls, varying by sex. Machine learning models highlighted specific microbial signatures for each sex, achieving high accuracy (area under the receiver operating characteristic curves of 0.91 for women and 0.72 for men). The key microbial taxa linked to MASLD included *Christensenella* and *Limosilactobacillus* in women and *Beduinibacterium* and *Anaerotruncus* in men. Functional profiling showed that MASLD patients had increased pathways for amine biosynthesis and amino acid degradation, while the controls exhibited enhanced fermentation pathways. These microbial features were associated with systemic inflammation, insulin resistance, and metabolite production linked to gut dysbiosis. **Conclusions**: The findings support the potential of gut microbiota signatures to be used as non-invasive indicators of MASLD and highlight sex-specific variations that could inform personalized diagnostic and therapeutic approaches.

## 1. Introduction

Metabolic dysfunction-associated steatotic liver disease (MASLD) is the most common form of chronic liver disease and is increasingly recognized as a significant public health challenge [1,2]. The disease includes both simple steatosis and its more severe form, metabolic dysfunction-associated steatohepatitis (MASH), which increases the risk of liver fibrosis, cirrhosis, hepatocarcinoma, and end-stage liver disease [1,3,4].

The prevalence of MASLD is expected to increase with rising rates of obesity, leading to concerns about its serious complications [4,5]. MASLD is usually overlooked and not recognized in primary care clinical practice. Moreover, the disease is considered a silent epidemic because its diagnosis is often delayed until it reaches an advanced degree [3]. Health professionals continue to highlight the need for reliable, non-invasive biomarkers that identify the disease in its early stages and, therefore, avoid its progression to irreversible stages [6,7]. 

The etiology of MASLD is considered multifactorial, with components like ethnicity, age and sex affecting its prevalence, development and progression rate [2,8]. Among these elements, the gut microbiota has also been reported to influence these aspects of the disease [6,9,10]. The microbiome has attracted interest in recent decades because of its direct effect on human health [11]. The gut microbiota has been proven to affect its host in many ways, such as producing metabolites and vitamins, maintaining intestinal barrier integrity, and regulating immune and inflammatory responses [9,11,12]. As a result of these functions, a disturbance or imbalance in the composition of the gut microbiota (dysbiosis) is likely to impact the host metabolism, making it a determinant factor in the development of several diseases [11,13]. 

Sex is a determinant factor of MASLD development and gut microbiota composition [6,7,8,9]. Moreover, regarding the composition of the gut microbiota, sex has proven to be a confounding factor in many gut microbiota analyses [7,8,10]. Sexual hormones are said to play a pivotal role in this observed sexual dimorphism [11]. Hence, it is recommended that all gut microbiota analyses are adjusted by sex. Other authors have stated the need to specifically study the sex differences regarding microbial composition in MASLD, and continue to elucidate its effect on the disease [7,8].

While studies have identified differences in the gut microbiota between MASLD patients and healthy individuals [12,13], more research is needed to characterize dysbiosis and explain how the gut microbiota influences MASLD [14,15]. 

Based on the previously stated, this study aims to elucidate the signature role of the gut microbiome in MASLD, identify non-invasive biomarkers and infer the altered metabolic pathways associated with the disease, considering sex differences in the study population.

## 2. Materials and Methods

### 2.1. Study Population and Design

This study analyzed baseline data from 98 MASLD patients and 45 healthy controls participating in the Fatty Liver in Obesity (FLiO) trial (NCT03183193), a randomized controlled nutritional intervention study conducted at the University of Navarra’s Centre for Nutrition Research (CIN). The study included adults (40–80 years) with MASLD who were overweight or obese (body mass index [BMI] 27.5–40 kg/m^2^) and diagnosed with hepatic steatosis via ultrasonography [16]. Controls had a normal weight (BMI < 25 kg/m^2^) and no hepatic steatosis. The exclusion criteria for the participants included the use of medications that could influence liver function, active infections, recent significant weight loss, alcohol abuse, existing liver or endocrine disorders, and the intake of any nutritional supplements (including prebiotics, probiotics and antioxidants) that could modify the composition of the gut microbiota. The study’s procedures were performed following the Declaration of Helsinki, with all participants providing informed consent. The present study presents baseline gut microbiota analyses, constituting a secondary outcome measurement for the FLiO trial. 

### 2.2. Anthropometric and Body Composition Data

Body composition, including visceral adipose tissue (VAT), was measured using dual-energy X-ray absorptiometry (DXA) under fasting conditions (Lunar iDXA, enCORE 14.5, Madison, WI, USA). BMI was calculated as weight divided by squared height (kg/m^2^).

### 2.3. Biochemical and Inflammatory Markers Determinations

Blood samples were collected after 8–10 h of fasting and stored at −80 °C. Biochemical markers, including glucose, insulin, cholesterol levels and triglycerides, were measured using commercial kits (Cobas 8000, Roche Diagnostics, Basel, Switzerland). Insulin resistance was assessed via the Homeostatic Model Assessment of Insulin Resistance (HOMA-IR) [17]. Liver fibrosis markers (CK-18 antigens M30 and M65) were quantified using ELISA kits (PEVIVA, Bromma, Sweden), while ferritin, C-reactive protein (CRP), leptin, chemerin, retinol-binding protein 4 (RBP4) and adiponectin were measured using ELISA and chemiluminescent assays (Demeditec; Kiel-Wellsee, Germany) in a Triturus autoanalyzer (Grifols, Barcelona, Spain). Leukocyte cell-derived chemotaxin-2 (LECT2) was quantified using specific kits (Biovendor LLC, Asheville, NC, USA). 

### 2.4. Hepatic Assessment

Liver status was assessed through ultrasonography, with MASLD diagnosis confirmed and graded (Siemens ACUSON S2000 and S3000, Siemens Healthineers, Erlangen, Germany). Aspartate aminotransferase (AST), alanine aminotransferase (ALT) and gamma-glutamyl transferase (GGT) were measured using automated analyzers (Cobas 8000, Roche Diagnostics, Rotkreuz, Switzerland). The hepatic fat content was determined via MRI (Siemens Aera 1.5T). Liver stiffness, an indicator of fibrosis, was assessed using Acoustic Radiation Force Elastography (ARFI) and Transient Elastography (TE). The Fatty Liver Index (FLI) was calculated based on triglycerides, BMI, waist circumference and GGT levels to estimate the likelihood of developing MASLD [18].

### 2.5. Fecal Sample Collection and Metagenomic Data

Fecal samples were collected using OMNIgene.GUT kits (DNA Genotek, Ottawa, ON, Canada). The samples were aliquoted and stored at −80 °C. Bacterial DNA was extracted using MagMAX™ CORE kits (Thermo Fisher Scientific, Whaltham, MA, USA) and the composition of the gut microbiota was analyzed through paired-end 16S rRNA gene sequencing using the Illumina protocol in a NovaSeq System (Illumina, San Diego, CA, USA). The Quantitative Insights Into Microbial Ecology (QIIME2) workflow was used for data processing, with DADA2 used for denoising and the BLAST and Ribosomal Data Project (RDP) databases used for taxonomic assignment. Quality was ensured with over 40,000 reads per sample [19,20].

### 2.6. Bioinformatic and Statistical Analysis

#### 2.6.1. Comparison of Baseline Characteristics

Baseline characteristics, including anthropometric data, hepatic parameters and biochemical markers, were compared using R Studio (Version 2023.09.0+463). Statistical significance was set at *p* < 0.05. Linear models and contrast analyses were performed using the emmeans evaluated group and sex differences, with results presented as mean and standard error of the mean (SE).

#### 2.6.2. Differential Abundance, Alpha and Beta Diversity and Metadata Association Analysis

MicrobiomeAnalyst software 2.0 was used for alpha and beta diversity analyses, with the Fisher, Shannon, ACE, Chao1 and Observed Alpha-diversity indexes calculated. Beta diversity was assessed using the Bray–Curtis distance and PERMANOVA tests. The differential abundance between the MASLD and control groups was analyzed using MaAsLin2 and ALDEX2 in R Studio, focusing on compositional data [21]. Low-abundance features were excluded (those with less than four counts in 80% of the participants) and feature associations with metadata were analyzed using Spearman correlations and visualized using the ComplexHeatmap package. Corrected *p*-values (*q*-values) < 0.05 were considered significant, using the Benjamini–Hochberg method for adjustment. Data were transformed using the Central Log Ratio (CLR).

#### 2.6.3. PICRUSt2 Functional Prediction and Feature Association Analyses

Phylogenetic Investigation of Communities by Reconstruction of Unobserved States 2.0 (PICRUSt2) software was used to predict functional pathways from raw 16S gene sequences. The Enzyme Classification (EC) numbers and MetaCyc pathway abundances were calculated. The differential pathway abundance was analyzed using MaAsLin2 with the Negative Binomial (NEGBIN) method. The associations between significant pathways and microbial features were examined using Spearman correlations, with significant *q*-values set at <0.05.

#### 2.6.4. Random Forest Classification Model

Random Forest (RF) classification models and machine learning techniques were built to identify possible biomarkers for MASLD in men and women. The RF models were trained using Python version 3 and the function RandomForestClassifier from the sklearn package, with a 5-fold cross-validation. The models used 2000 estimators and no maximum depth. The model for women used weights of 4 for the negative class and 1 for the positive class, while the model for male subjects used no class weights. The input data were the CLR-transformed genus counts of the genii present in at least 80% of the participants. The Gini importance of each feature was extracted from the models to obtain potential biomarkers.

## 3. Results

### 3.1. General Characteristics of the Study Population

The control and MASLD groups presented similar characteristics in terms of age and sex [22]. The study population was separated by sex, given the strong effect of this factor on the composition of the gut microbiota [6,8,11,23]. Most of the anthropometric parameters, body composition parameters, hepatic parameters, and inflammatory and biochemical measurements were significantly different in the controls compared to MASLD individuals, regardless of sex. 

MASLD individuals generally possessed measurements suggesting a worse metabolic and hepatic status, glucose metabolism and inflammatory state. In men, transaminases, M65, CRP and chemerin did not differ between groups. In women, the AST, GGT, M65, ferritin and LDL cholesterol levels did not differ between the controls and MASLD volunteers (Table 1). Moreover, some evaluated parameters significantly differed in men and women when comparing the controls and MASLD groups. Specifically, the control group showed significant differences in weight, waist, FLI index, hepatic volume, transaminases, ferritin, RBP4, leptin, adiponectin and HDL-c levels between men and women. Furthermore, in the MASLD group, there were differences in weight, waist, visceral adipose tissue (VAT), FLI index, hepatic volume, transaminases, ferritin, RBP4, leptin, glycated hemoglobin, HDL-c and triglyceride levels between men and women (Table 1). 

### 3.2. Alpha and Beta-Diversity

The groups’ alpha diversity was evaluated using the Shannon, Chao1, Fisher, ACE and observed indexes. In both women and men, MASLD individuals had a significantly lower alpha diversity than the controls in all the evaluated indexes (Figure 1a,b). Furthermore, the beta diversity was significantly different between the MASLD and control groups in both sexes (Figure 1c,d).

### 3.3. Differential Abundance Analyses

Differential abundance analysis was performed using MaAsLin2 and later corroborated by using ALDEX2. Given its compositional nature, both statistical tools are specifically designed for this type of analysis regarding the composition of the gut microbiota. 

In women, MaAsLin2 analysis found 17 species, 16 genera and 3 families that significantly differed between the MASLD and control groups (Appendix A). Among those, 11 species, 12 genera and 3 families were also found in the ALDEx2 analysis (Appendix A). The genera *Anaerutruncus*, *Anaerofilum*, *Christensenella*, *Harryflintia*, *Intestinimonas*, *Paludicola*, *Peptococcus*, *Pseudoflavonifractor* and *Sporobacter* were significantly depleted in MASLD subjects in both MaAsLin2 and ALDEx2. On the other hand, the genera *Limosilactobacillus*, *Phocaeicola* and *Tyzzerella* were increased considerably in MASLD subjects in both differential analyses. Moreover, in the MaAsLin2 and ALDEX2 analyses, the *Desulfobacteriaceae*, *Peptococcaceae* and *Christensenellaceae* families were significantly depleted in MASLD women. Boxplot comparisons and heat trees of these significant taxa can be found in Appendix A.

In men, the MaAsLin2 analysis determined 12 species, 8 genera and 8 families that significantly differed between the MASLD and control groups (Appendix A). Among those, eight species and all genera and families were also found in the ALDEx2 analysis (Appendix A). Among the genera significantly depleted in MASLD men in both differential analyses were *Acholeplasma*, *Beduinibacterium*, *Harryflintia*, *Ihubacter*, *Ligilactobacillus*, *Peptococcus* and *Sporobacter*. On the other hand, the *Parabacteroides* abundance was increased in MASLD men according to the MaAsLin2 and ALDEx2 analyses. All the families significantly different from the MaAsLin2 analysis were also significant in the ALDEx2 analysis. Among the families depleted in MASLD subjects were *Peptococcaceae* 1, *Acholeplasmataceae*, *Puniceicoccaceae*, *Synergistaceae* and *Oxalobacteraceae*. On the other hand, the families *Porphyromonadaceae*, *Streptococcaceae* and *Anaeroplasmataceae* were significantly increased in MASLD men. Boxplot comparisons and heat trees of these significant taxa can be found in Appendix A.

*Peptococcus simiae*, *Sporobacter termitidis* and *Harryflintia acetispora* were significantly depleted in men and women with MASLD in the MaAsLin2 and ALDEx2 analyses. Moreover, *Parabacteroides gordonii* was also depleted in men and women with MASLD, according to the ALDEx2 results. Similarly, the genera *Peptoccocus*, *Sporobacter* and *Harryflintia* were also significantly depleted in men and women with MASLD. According to the MaAsLin2 and ALDEx2 analyses, the *Peptococcaceae* 1 family was also depleted in both sexes.

### 3.4. Metadata Associations

Associations between the significantly different microbial features and the subjects’ clinical metadata were evaluated (Figure 2). The clinical metadata variables evaluated included body composition, biochemical and inflammatory parameters, and glycemic and lipid profile measurements.

In women, the genera *Christensenella*, *Massiliprevotella*, *Peptococcus*, *Harryflintia*, *Intestinimonas*, *Sporobacter* and *Anaerotruncus* were among the significant features that negatively correlated with the clinical metadata linked to the disease, such as a higher BMI, liver fat percentage and FLI index. Moreover, they were also negatively associated with inflammatory and glycemic profile parameters, such as CRP, leptin, the HOMA index and glycated hemoglobin. Additionally, most of these genera are positively associated with adiponectin levels. Contrarily, the genera *Tyzerella*, *Limosilactobacillus*, *Phocaeicola* and *Paramuribaculum* positively correlated with the previously mentioned variables, related to an overall worse metabolic state (Figure 2a). The exact *q*-values for these associations can be found in Appendix A. 

In men, the *Parabacteroides* genus positively correlated with typical MASLD-associated variables, such as BMI, liver fat percentage and the FLI index, as well as several inflammatory markers, including LECT2, RBP4 and leptin. The genera *Beduinibacterium*, *Sporobacter*, *Harryflintia*, *Peptococcus* and *Ihubacter* negatively correlated with the previously mentioned parameters and were linked to a healthier metabolic state (Figure 2b). The exact *q*-values for these associations can be found in Appendix A. The genera *Peptococcus*, *Sporobacter* and *Harryflintia* were negatively associated with BMI, liver fat, the FLI index and leptin in both men and women.

### 3.5. Predicted Functional Analyses

Predicted functional analysis was performed using PICRUSt2 to evaluate which metabolic pathways are altered, enhanced, or diminished in controls and MASLD subjects based on the composition of their gut microbiota. Control women seemed to have increased amino acid biosynthesis, fermentation and nucleotide degradation pathways. Opposingly, MASLD women had enhanced amines and siderophore biosynthesis, amino acids degradation, aromatic compound degradation and sugar derivative degradation pathways according to the PICRUSt2 analyses (Appendix A). 

Moreover, in the men’s sample, the controls appeared to have enhanced fermentation, amino acid, aromatic compound, fatty acid biosynthesis and nucleotide degradation pathways. Contrarily, men with MASLD seemed to have enhanced amino acid and aromatic compound degradation, amine, vitamin and siderophore biosynthesis, and generation of precursor metabolites and energy pathways (Appendix A). Among the paths enhanced in men and women with MASLD were those involved in aromatic compound and amino acid degradation, and amine and siderophore biosynthesis. On the other hand, among the pathways enhanced in healthy controls from both sexes were amino acid biosynthesis, fermentation and nucleotide degradation pathways. Identical behaviors between men and women were found regarding the common pathways when comparing the MASLD and control groups (Appendix A). 

The association of the metabolic pathways with significant features from the differential abundance analyses was evaluated in women and men separately, and can be seen in Figure 3a and Figure 3b, respectively. The exact *q*-values from the associations presented in Figure 3 can be found in Appendix A.

### 3.6. Machine Learning Selection of Potential Biomarkers

RF machine learning models were employed to identify critical discriminatory features of the disease. The genus taxonomic level was used to select said features. When constructing RF models to obtain possible biomarkers for the disease, taxa were ordered according to the Gini importance criterion in the model. By selecting those features that could better classify subjects, the most important and relevant features for the disease were chosen in men and women separately. 

In women, 21 taxa were responsible for up to 80% of the classification obtained from the RF model (Figure 4a). In this case, the capacity of the genera *Christensenella* and *Limosilactobacillus* to classify the subjects was up to 40%. Both genera also appeared among the features that differed significantly in the differential abundance analysis between control and MASLD women. Concretely, *Christensenella*’s abundance was enhanced in controls, while *Limosilactobacillus*’s abundance was enhanced in MASLD women. Both RF models successfully classified subjects in men and women (area under the receiving operating curve [AUC] of 0.72 and 0.91, respectively). Several RF models’ taxa also appeared statistically significant in the differential abundance analyses. 

Men had 11 taxa responsible for up to 80% of the classification obtained from the RF model into MASLD or control groups (Figure 4b). Among these taxa, the *Beduinibacterium* and *Anaerotruncus* genera had classification capacities of more than 40%. *Beduinibacterium* was a genus found in the differential abundance analysis and was depleted in MASLD individuals. On the other hand, *Anaerotruncus*’s abundance was enhanced. 

## 4. Discussion

The baseline characteristics of the MASLD and control subjects differed as expected, with MASLD individuals showing a worse metabolic profile, characterized by a higher BMI, waist measurements, body fat and visceral fat. These individuals also exhibited poorer glycemic control, higher insulin resistance and an increased inflammatory state, confirming the link between metabolic syndrome, inflammation and MASLD development [1,3,24].

Regardless of MASLD status, men showed higher FLI and transaminase levels than women. Men also had higher visceral adipose tissue, waist measurements, RBP4, ferritin and leptin levels, all linked to the development of MASLD. This suggests a potential explanation for men’s increased susceptibility to liver disease [3,6]. The role of sex in disease progression should be further investigated and future nutritional interventions might benefit from sex-specific adjustments [6,8].

Gut microbiota analyses revealed significant differences between MASLD and controls in both sexes. Alpha diversity was significantly higher in the control groups, aligning with other studies that link increased diversity to better metabolic health, improved glucose homeostasis and lower levels of obesity [12]. Beta diversity also differed between MASLD and controls, highlighting the substantial shifts in gut microbiota associated with disease status [25]. These changes were observed in both men and women, suggesting that MASLD distinctly alters microbial composition regardless of sex. 

The differential abundance analyses performed showed that the majority of the significantly altered taxa were sex dependent. Only three genera, *Harryflintia*, *Peptococcus* and *Sporobacter*, were commonly different between the MASLD and control groups in both men and women. This suggests that while MASLD alters microbial composition, the specific changes differ based on sex. The robustness of these results is further validated by the consistency between the two analytical methods used, as both showed similar patterns despite methodological differences [21]. 

Several identified microbial taxa were associated with crucial metabolic and inflammatory markers. The taxa enriched in MASLD individuals were linked to indicators of poor metabolic health, insulin resistance and systemic inflammation, which are hallmarks of the disease [3,12]. Conversely, the depleted taxa in MASLD subjects correlated negatively with these adverse markers, suggesting that these microbes may play a protective role [12].

One notable genus, *Christensenella*, was significantly depleted in MASLD women and appeared in both differential abundance analyses and the RF model. *Christensenella* was also less abundant in men in MASLD individuals, but the results did not reach statistical significance. This genus has been extensively discussed in the literature for its potential role as a probiotic associated with improved body composition and metabolic health [26]. Interestingly, *Beduinibacterium*, the genus with the highest importance in the men’s RF model, has up to 90% phylogenetical identity with *Christensenella* [27].

The genus *Tyzzerella* was significantly increased in MASLD women and emerged as the third most important feature in the RF model. This genus was also strongly associated with inflammation. Other studies have reported similar findings linking *Tyzzerella* to inflammatory diets and increased visceral adipose tissue [28]. The association of *Tyzzerella* with visceral adipose tissue and liver function indicators, particularly in women, could explain why it was significant only in female MASLD subjects.

In men, the genus *Anaerotruncus* was significantly enriched in MASLD individuals, emerging as the second most important feature in the RF model. *Anaerotruncus* has been associated with obesity, liver steatosis and metabolic disturbances in other studies, particularly in animal models [29]. This bacterium may contribute to MASLD development through its involvement in nutrient-related metabolic pathways.

Common genera across both sexes, such as *Harryflintia*, *Peptococcus* and *Sporobacter*, decreased in MASLD subjects. *Harryflintia* is an acetate producer, a short-chain fatty acid (SCFA) linked to improved metabolic health and inflammation reduction [30]. Moreover, increased *Peptococcus* abundance is a protective factor for hepatic steatosis and an increased abundance of the *Peptococcaceae* family is found after weight loss [31]. Furthermore, *Sporobacter* has been linked to longevity and health span [32], and negatively associated with hypertension [33]. 

Several identified taxa were also associated with pathways linked to SCFA production and amino acid degradation. SCFA-producing pathways, such as pyruvate fermentation, were increased in control subjects and positively related to beneficial bacteria [34]. Conversely, amino acid degradation pathways, which produce harmful metabolites like ammonia and polyamines, were elevated in MASLD individuals, particularly in women [34]. These pathways contribute to insulin resistance, obesity and MASLD progression, reinforcing the connection between gut microbial composition and disease pathogenesis.

One of the limitations of the currently published literature about the gut microbiota in the context of health and disease is the need for more functional knowledge. While differential abundance and change analyses can provide helpful information, mechanisms must be inferred to establish the possible effect of microbial composition on actual host health. This study addressed this limitation using PICRUSt2 functional analysis, which has considerable benefits compared to other functional prediction software [35,36]. Nevertheless, PICRUSt2 relies on inference rather than the direct measurement of genes and metabolic pathways. Future studies could reduce this limitation by using shotgun metagenomics, which enables a more accurate and comprehensive assessment of microbial functionality by sequencing the entire microbial genome.

Another limitation when studying the gut microbiota, in general, is the lack of standardization in the study’s methodologies and the scarce scientific information there is about many taxa, which include some of those found in this research. This makes the discussion and elucidation of the results quite difficult. Nevertheless, the study partially mended this by corroborating the results found in differential abundance analyses with two different methodologies and by later using machine learning classification technologies, which ratified the significant taxa involved in the disease [21,37]. Moreover, as mentioned previously, the results and the differential analysis had plausible biological explanations and contributed to elucidating the role of microbial composition in MASLD. Finally, it is worth mentioning that this trial was also performed in the Spanish population. Therefore, its results might vary from other samples, considering that ethnic and genetic differences influence microbial composition. As in other gut microbiota research, ratifying these results in other cohorts is necessary.

On the other hand, this study possesses many positive traits and strengths as well, such as the characterization of the study subjects, which includes an extensive hepatic and body composition evaluation, the determination of several biochemical and inflammatory markers involved in the disease, and the use of high-throughput sequencing technologies. Moreover, the novelty of the article’s topic, the need to evaluate it and the use of several bioinformatic tools and validated machine learning techniques make this study particularly interesting. Additionally, finding potential non-invasive biomarkers for a silent disease, such as MASLD, makes this research attractive, especially now in the era of precision and integrative medicine.

Moreover, these findings could be employed in therapeutical and practical applications for managing MASLD. The finding of sex-specific microbial signatures underscores the need for a precise and personalized approach when treating and diagnosing the disease. The microbial features found, particularly those associated with inflammation, insulin resistance and detrimental metabolite production, could serve as non-invasive biomarkers and screening tools for MASLD. Additionally, the functional analysis provides opportunities to target dysregulated metabolic pathways, such as amine biosynthesis and amino acid degradation, which could open new avenues for therapeutic interventions.

The results found in this study are intrinsically involved with precision nutrition and precision medicine, as they present the possibility of microbiome-targeted therapies that could aim to either change the bacterial composition or modulate specific pathways through dietary strategies, the intake of functional foods, and probiotic and postbiotic supplementation. Finally, this research supports a shift towards more targeted interventions and sex-specific therapeutic strategies, paving the way for improved MASLD management and patient outcomes.

## 5. Conclusions

Microbial features and their functional profiling could elucidate the effect of the gut microbiota on the disease, and they could also function as noninvasive markers for MASLD. Additionally, sex plays an essential role in gut microbiota composition and several of the microbiome’s characteristic features are sex dependent. These findings provide valuable insights for personalized nutrition and precision medicine, suggesting that the composition of the gut microbiota could be a target for managing metabolic diseases, such as MASLD.

## Figures and Tables

**Figure 1 nutrients-16-04198-f001:**
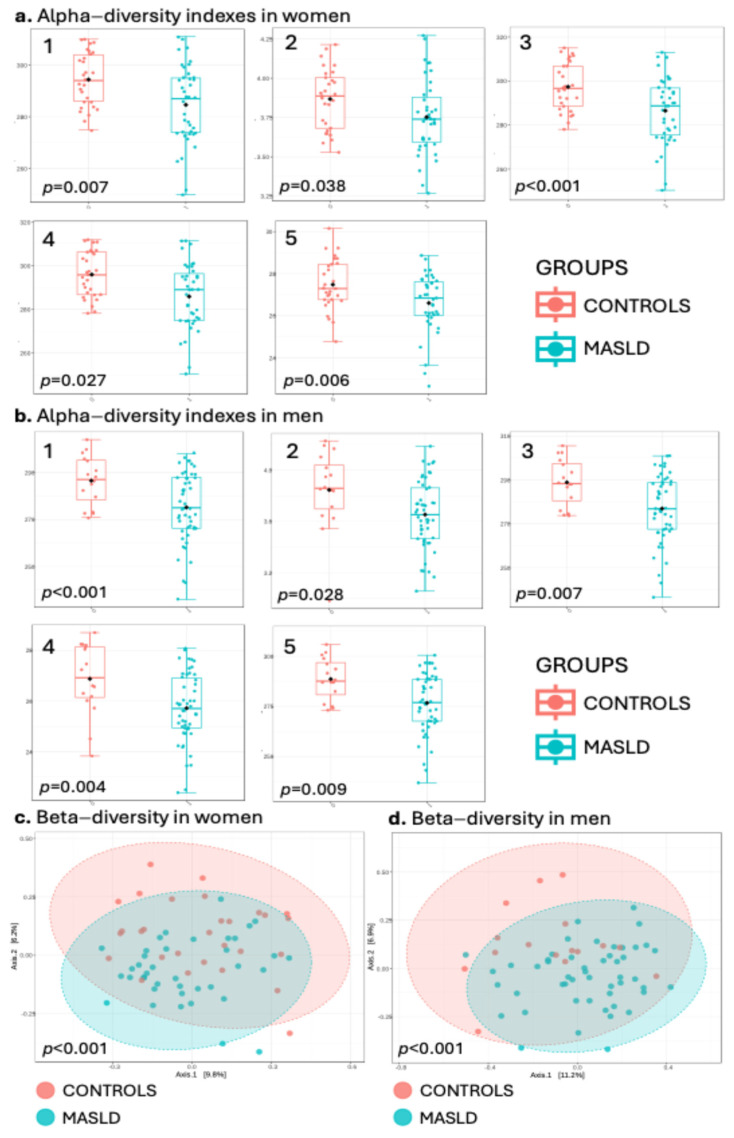
Alpha-diversity index comparison between the control and MASLD groups in (**a**) women and (**b**) men. The indexes shown are as follows: 1. Observed; 2. Shannon; 3. Chao1; 4. Fisher; 5. ACE. Beta-diversity comparison between control and MASLD groups in (**c**) women and (**d**) men.

**Figure 2 nutrients-16-04198-f002:**
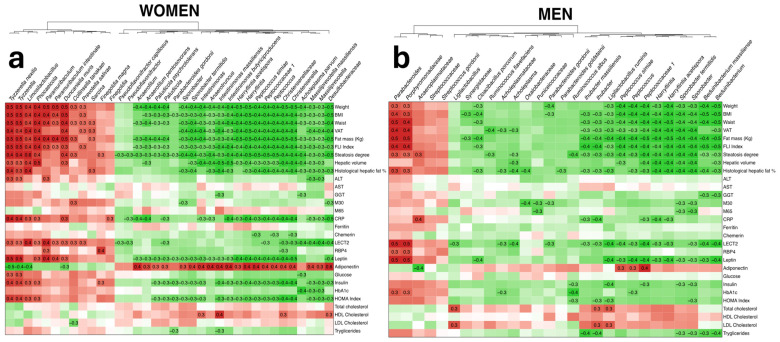
Metadata association with significant microbial features in (**a**) women and (**b**) men. Negative associations are presented in green and positive associations are presented in red. The intensity of the color represents the degree of the association, with a higher Rho marked as more intense. Significant associations with *q*-values less than 0.05 are marked with their corresponding Rho values.

**Figure 3 nutrients-16-04198-f003:**
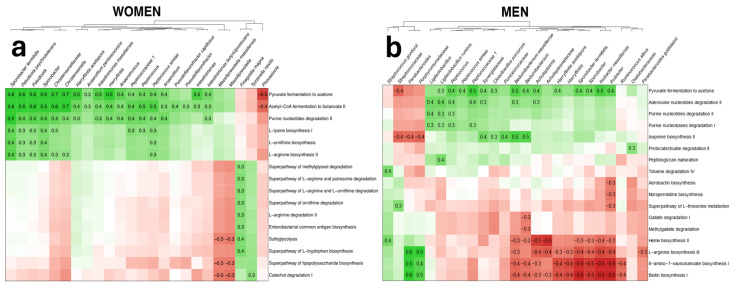
Associations between bacterial features and altered metabolic pathways in (**a**) women and (**b**) men. Negative associations are presented in red and positive associations are presented in green. The intensity of the color represents the degree of the associations, with a higher Rho marked as more intense. Significant associations with *q*-values less than 0.05 are marked with their corresponding Rho values.

**Figure 4 nutrients-16-04198-f004:**
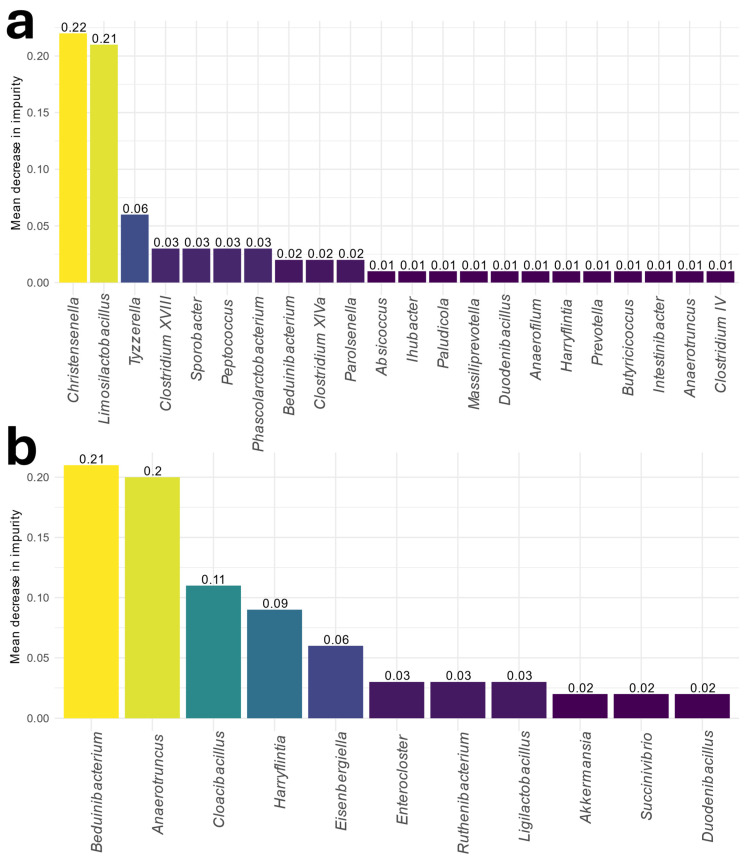
Feature importance according to the Gini criterion from the RF models in (**a**) women and (**b**) men.

**Table 1 nutrients-16-04198-t001:** General characteristics and comparisons between groups and sex of the study subjects.

	Men (*n* = 66)			Women (*n* = 68)				
	Controls (*n* = 16)	Masld (*n* = 43)	*p*-Value Controls vs. Masld	Controls (*n* = 29)	Masld (*n* = 55)	*p*-Value Controls vs. Masld	*p*-Value Controls Men vs. Women	*p*-Value Masld Men vs. Women
	Anthropometry and body composition assessment							
Weight (kg)	77.45 (2.70)	101.25 (1.50)	<0.001	59.083 (2.011)	85.359 (1.735)	<0.001	<0.001	<0.001
BMI (kg/m^2^)	24.29 (0.82)	33.91 (0.46)	<0.001	22.71 (0.61)	32.78 (0.53)	<0.001	0.129	0.108
Waist (cm)	85.84 (1.99)	113.19 (1.10)	<0.001	74.88 (1.48)	102.94 (1.27)	<0.001	<0.001	<0.001
Body Fat (%)	17.47 (1.82)	38.05 (1.01)	<0.001	20.31 (1.35)	39. 38 (1.16)	<0.001	0.212	0.391
VAT (kg)	0.75 (0.17)	2.84 (0.09)	<0.001	0.36 (0.13)	1.53 (0.11)	<0.001	0.080	<0.001
	Hepatic parameters							
FLI Index	28.70 (4.14)	86.45 (2.29)	<0.001	10.54 (3.07)	67.54 (2.65)	<0.001	0.001	<0.001
Histological Liver Fat (%)	3.38 (1.38)	11.52 (0.77)	<0.001	2.86 (1.04)	9.50 (0.89)	<0.001	0.761	0.091
Hepatic volume (mL)	1500.37 (82.16)	1934.95 (46.95)	<0.001	1204.89 (62.11)	1582.02 (53.31)	<0.001	0.005	<0.001
ARFI stiffness (m/s)	1.38 (0.16)	1.75 (0.08)	0.036	1.23 (0.116)	2.05 (0.01)	<0.001	0.457	0.051
TE stiffness (kPa)	4.56 (0.44)	5.27 (0.23)	0.134	4.11 (0.31)	4.67 (0.28)	0.174	0.382	0.102
ALT (IU/L)	33.1 (3.90)	36.34 (2.16)	0.468	15.08 (2.89)	29.74 (2.5)	<0.001	<0.001	0.048
AST (IU/L)	30.61 (2.42)	26.34 (1.34)	0.126	19.77 (1.80)	22.61 (1.55)	0.235	<0.001	0.072
GGT (IU/L)	45.12 (7.36)	44.94 (4.08)	0.983	17.41 (5.46)	27.00 (4.71)	0.187	0.003	0.005
	Inflammatory markers							
M30 (U/L)	36.59 (16.15)	93.82 (8.67)	0.002	43.09 (12.04)	96.53 (10.14)	0.001	0.747	0.840
M65 (U/L)	110.22 (23.06)	151.92 (12.39)	0.114	117.19 (17.19)	138.23 (14.30)	0.349	0.809	0.471
CRP (mg/dL)	0.08 (0.28)	0.34 (0.15)	0.418	0.09 (0.20)	0.69 (0.17)	0.030	0.962	0.134
Ferritin (ng/mL)	121.65 (23.02)	204.41 (12.77)	0.002	50.83 (17.10)	69.05 (14.74)	0.421	0.015	<0.001
Chemerin (ng/mL)	175.81 (13.76)	204.40 (7.63)	0.071	192.41 (10.22)	221.23 (8.81)	0.035	0.335	0.151
LECT2 (ng/mL)	24.43 (2.50)	41.80 (1.38)	<0.001	23.50 (1.86)	41.49 (1.60)	<0.001	0.765	0.883
RBP4 (mg/L)	30.12 (2.12)	38.97 (1.17)	<0.001	24.79 (1.57)	31.18 (1.36)	0.003	0.046	<0.001
Leptin (ng/mL)	5.73 (5.11)	22.18 (2.74)	0.005	18.49 (3.68)	57.56 (3.17)	<0.001	0.045	<0.001
Adiponectin (µg/mL)	11.13 (0.88)	6.61 (0.49)	<0.001	15.61 (0.66)	6.82 (0.56)	<0.001	<0.001	0.787
	Biochemical measurements							
Glucose (mg/dL)	93.60 (3.41)	105.10 (1.93)	0.004	89.15 (2.53)	99.61 (2.18)	0.002	0.298	0.062
HbA1c (%)	5.01 (0.16)	5.89 (0.09)	<0.001	5.20 (0.12)	5.60 (0.10)	0.014	0.337	0.036
HOMA-IR	0.85 (0.51)	4.65 (0.28)	<0.001	0.99 (0.37)	4.11 (0.32)	<0.001	0.828	0.217
Total cholesterol (mg/dL)	219.50 (9.57)	193.15 (5.31)	0.017	217.17 (7.11)	197.23 (6.13)	0.036	0.846	0.616
LDL-cholesterol (mg/dL)	139.78 (8.33)	116.42 (4.66)	0.016	130.06 (6.18)	116.77 (5.33)	0.106	0.351	0.960
HDL-cholesterol (mg/dL)	62.12 (3.53)	48.44 (1.96)	0.001	70.87 (2.62)	59.05 (2.26)	0.001	0.049	<0.001
Triglycerides (mg/dL)	87.93 (13.04)	145.51 (7.23)	<0.001	81.17 (9.69)	107.05 (8.35)	0.045	0.678	<0.001

Data represent the mean (SD). The *p*-values for controls vs. MASLD show the comparison between groups of the same sex. The *p*-values for men vs. women show a comparison between sexes within the same group. Abbreviations: ALT, Alanine aminotransferase; ARFI, Acoustic Radiation Force elastography; AST, Aspartate aminotransferase; BMI, Body Mass Index; CRP, C-reactive protein; FLI, Fatty Liver Index; GGT, Gamma-glutamyl transferase; HbA1c; Glycated Hemoglobin; HDL, High-Density Lipoprotein; HOMA-IR, Homeostatic Model Assessment for Insulin Resistance; LECT2, leukocyte cell-derived chemotaxin-2; LDL, Low-Density Lipoprotein; RBP4, Retinol-Binding Protein 4; TE, transient elastography; VAT, Visceral Adipose Tissue. M30 and M65 are cytokeratin 18 (CK-18) antigens.

## Data Availability

The original contributions presented in the study are included in the article/Appendix A, further inquiries can be directed to the corresponding author.

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
