# Peer review of "Sex-Dependent Gut Microbiota Features and Functional Signatures in Metabolic Disfunction-Associated Steatotic Liver Disease"

_nutrients, 2024, doi:10.3390/nu16234198_

Round 1

Reviewer 1 Report

Comments and Suggestions for Authors

Dear Authors,

This study was conducted to sex-dependent gut microbiota features and functional signatures in metabolic disfunction-associated steatotic liver disease. This study focused on a topic that has not been widely covered in the field of gene, molecular, biology, and public health science. I believe this was excellent issue in field of medicine section, too.

Abstract: well-written

Line 27: FLiO à Fatty Liver in Obesity

Line 31: Please change AUROC to full name.

Please add the results of statistical exact p-value in each variables in Results section.

Please sort alphabetically in Key-words.

1. Introduction

Please add some paragraphs that background or literature review between gut microbiota features and functional signature based on sex-dependent. The introduction section context was too short.

2. Method: well-written

Line 71: FLiO (Fatty Liver in Obesity) trial à Fatty Liver in Obesity (FLiO) trial

Line 74: (BMI 27.5-40 kg/m²) à (body mass index [BMI] 27.5-40 kg/m²)

Abbreviations are usually defined at the first use

Line 86: Body mass index (BMI) à BMI

3. Results

Well-describe

4. Discussion

You should add strengths, limitations, application in field of this study.

Furthermore, checking by the iThenticate system, the plagiarism rate was 31% (quotes included and bibliography excluded). I think it is not acceptable plagiarism rate. Please reduce the plagiarism rate.

Reviewer 2 Report

Comments and Suggestions for Authors

1. Authors should modify the structure of the manuscript. In the results section under a given subsection, the obtained results should be cited directly below the comment. Placing all results after their comments makes the manuscript less readable.

2. Figures 2 and 3 show a large amount of data, which makes it difficult to read; the authors should try to increase its resolution.

3. Including more discussion on therapeutic potential (e.g., probiotics) could enhance the manuscript's relevance in the discussion section.

4. Some sections referencing supplementary materials disrupt the flow and require better integration.

5. The conclusions section requires extension; I suggest the authors address potential further applications or limitations of the developed model and the need to conduct additional research.

Comments on the Quality of English Language

The manuscript requires minor stylistic and linguistic corrections.
